# Tools and Metrics for Species Prioritization for Conservation Planning and Action: Case Studies for Antelopes and Small Mammals

Thomas E. Lacher, Jr. [1,2,3,6,*] , David Mallon [4,7] , Rosalind J. Kennerley [5,6] , Claire Relton [5,6] and Richard P. Young [5,6]

1   Department of Ecology and Conservation Biology, Texas A&M University, College Station, TX 77843-2258, USA
2   Re:wild, P.O. Box 129, Austin, TX 78767, USA
3   Instituto Neotropical: Pesquisa e Conservação, Rua Purus 33, Curitiba 82520-750, Paraná, Brazil
4   Department of Natural Sciences, Manchester Metropolitan University, Chester St., Manchester M1 5GD, UK
5   Durrell Wildlife Conservation Trust, Trinity JE3 5BP, Jersey, Channel Islands, UK
6   IUCN Species Survival Commission, Small Mammal Specialist Group
7   IUCN Species Survival Commission, Antelope Specialist Group
*   Correspondence: tlacher@tamu.edu

**Abstract:** Given the scale of the current biodiversity loss, setting conservation priorities is essential to direct scarce resources to where they will be most effective. Many prioritization schemes have been described by using a wide range of criteria that vary across taxonomic groups, spatial scales, and ecological, socio-economic, and governance contexts. Currently, there is no single prioritization process applicable to all situations, nor is there a list of agreed metrics. The IUCN SSC Antelope Specialist Group and the Small Mammal Specialist Group recently performed species prioritization exercises based on a similar approach. The variables used included biological, socio-political, and feasibility criteria. The two exercises contained both common and some unique variables, arranged in a matrix for the target species (29 threatened antelopes and 19 critically endangered Mexican small mammals, respectively). The ASG framework provided a global summary of the antelope priorities, which can be updated and adapted to the national level. The SMSG matrix was applied in a regional workshop to select species for which the likelihood of implementing conservation actions was high and led to conservation action plans being developed for six species. The framework we jointly developed in theory can be applied to other taxa, certainly all mammals and perhaps most vertebrates.

**Keywords:** IUCN; Red List; extinction risk; conservation priorities; assessments

## 1. Introduction

Global biodiversity is under threat from a multitude of anthropogenic factors [1], and the current crisis has been referred to as a global annihilation and the sixth mass extinction [2]. This crisis has stressed the immediacy of conservation action [3–5] to stop and reverse these declines. The necessary actions, however, will depend upon the threats' severity as well as the likelihood and feasibility of a successful intervention [3,6]. Biological, social, economic, and governance factors are all in play simultaneously, and a failure to account for them can render even a well-funded conservation action a failure. There is an urgent need to prioritize species for conservation action and to focus and make the best use of scarce resources where they are most needed; however, there is no broad consensus on how best to structure this prioritization.

Species selection and prioritization for conservation action are influenced by a range of rationales and perspectives, such as the value and/or vulnerability of a species, the directives, and priorities of the conservation organization responsible for implementation [7], the ease at which funding can be secured for a given flagships species [8], or the cultural

value of the species to local communities [9]. However, selecting species based solely on one criterion, such as their Red List (or threat) status, may result in the oversimplification of a complex situation [10] or may overlook species that have not been recently reviewed or that are data deficient [11].

In addition to the species' need for protection, as defined by their Red List category and feasibility of conservation action, Arponen [12] advocates for the use of biologically relevant criteria for species-level prioritization, such as species uniqueness, evolutionary potential, and ecological function. There are reviews of mixed results describing the value of planning conservation based on the selection of umbrella, flagship, or indicator species [13] and if their protection serves to conserve co-existing species defined by criteria, such as their uniqueness, charisma, sensitivity to disturbance, and proportion of co-occurring species [8,14–18].

The Conservation Needs Assessment methodology, created by Amphibian Ark, uses the available knowledge of wild amphibians to identify those with the most urgent conservation needs to develop national lists of species prioritized for conservation [19]. The Conservation Needs Assessment links to the IUCN Red List Assessment and was originally developed during an amphibian conservation workshop in 2006, facilitated by the Conservation Breeding Specialist Group. The Conservation Needs Assessment process considers the IUCN Red List category and the evolutionary distinctiveness (EDGE) score from the Evolutionarily Distinct and Globally Threatened program. In addition, the process makes use of a set of weighted questions on topics, including the species' status, threats, and recovery; significance (cultural, scientific, biological, and socio-economic); and previous and potential conservation actions, to create a decision tree and generate prioritized lists of species recommended for conservation and a range of suggested and prioritized conservation actions [19]. Provided that there is a broad range of experts available, there is a potential to adapt and use this tool for other taxonomic groups.

Similarly, a UNEP/GEF project aiming to conserve wild plant species used a set of criteria and a scoring system to select priority species [10]. This approach was an adaptation of that described in [20], who proposed an approach of using scientific criteria associated with scored (weighted) indicators to establish priorities. These criteria were grouped into five categories of indicators, including threat (extinction risk), conservation (existence of conservation plans), genetic (genetic potential and importance), economic (economic importance) and utilization (social importance) [20]. Other pragmatic considerations included: (1) The probability of project success and sustainability; (2) the financial feasibility of implementation; (3) the taxonomic certainty of the species; (4) the technical feasibility of species monitoring; and (5) other biological traits [10]. The specific variables considered for wild crop relatives included the IUCN Red List category and associated threats, endemicity, and the genetic information relevant to cultivated and wild forms.

It may be challenging and unrealistic to suggest that a standardized set of criteria can be used for selecting species across all conservation contexts, taxonomic groups, and spatial scales. Le Berre et al. [21] assessed 40 studies and 24 varying scoring and rule-based methods for species prioritization to assist practitioners in choosing appropriate methods. Specific criteria were not defined, and their selection and emphasis should suit the specific management context in which the project exists. The Standardized Index of Vulnerability and Value Assessment (SIVVA), described by Reece and Noss [22], is a technique designed specifically for coastal species threatened by sea-level rises. This framework utilizes a flexible scoring system and includes metrics for both conservation value and vulnerability, which can be emphasized or deemphasized according to the user's objectives [22]. Regardless of the criteria or methods used, uncertainty exists, especially when only poor-quality data are available [12]. However, if explicit and defensible objectives are defined, and choices are scientifically justified, users will be able to select criteria that are appropriate for their purposes [21].

## 2. Materials and Method

### 2.1. IUCN and Species Prioritization

The IUCN Red List of Threatened Species is the globally recognized tool for assessing an extinction risk and is widely used by nearly all major conservation conventions and treaties [23–25]. The Red List Assessments themselves are static, however, and may only be revised in 5- to 10-year time frames. IUCN has recently developed a more dynamic framework, the Assess, Plan, and Act framework, to integrate assessments, planning, and conservation action. Assess–Plan–Act is defined as the Species Conservation Cycle (https://www.reversethered.org/species-conservation-cycle) and is a key component of the Reverse the Red campaign of SSC. Assess is the well-recognized process of conducting IUCN Red List assessments and publishing the spatial, population, threat data, and other supporting information on the IUCN Red List of Threatened Species website (https://www.reversethered.org/species-conservation-cycle, accessed on 25 July 2022). Assessed species must also be reassessed, usually within five to ten years, which allows for tracking changes in the status by species, larger clade, or region through the calculation of the Red List Index. These trend line data points are also widely used in international conservation conventions and the development of National Reports and National Biodiversity Strategies and Action Plans (NBSAPS); [26]. However, the often-long interval between reassessments can be slower than the speed of population declines and changes in the intensity of threats and drivers for many species.

Assess–Plan–Act adds the additional components of planning and conservation action to the cycle. The planning process will bring together multiple stakeholders and biological and social-economic expertise from the public, civil society, and governments to develop evidence-based conservation plans. Among the published prioritization examples, the most relevant to our assessments [27] addressed the Venezuelan species of birds and used weighted variables, including the IUCN Red List category and an assessment of extinction risk, endemicity, taxonomic uniqueness, and the likelihood of public support for conservation action. Nevertheless, there is currently no list of metrics, nor a consistent process, for prioritizing species, and what is needed is the development of a suite of tools and metrics for each situation or taxonomic group to assist in the planning process and arrive at the best priorities for action.

Two IUCN SSC specialist groups, the Antelope Specialist Group (ASG) and the Small Mammal Specialist Group (SMSG), recently performed species prioritization exercises for their respective groups. Firstly, the ASG compiled some broad species prioritization criteria following the completion of the Antelope Survey and Regional Action Plan [28] and developed these further after the latest Global Mammal Assessment (2016–2018). Secondly, the SMSG undertook a species prioritization exercise for species in Mexico, which is an important key region for the group. A recent global analysis of all rodents and eulipotyphlans demonstrated that Mexico has high aggregations of both globally threatened (those listed as either critically endangered (CR), endangered (EN), or vulnerable (VU)) and data deficient (DD) species of rodents, shrews, and moles [29]. One of the main motivations for undertaking the analysis was for the results to be used as a starting point to stimulate local engagement and, in doing so, drive locally run expert workshops to explore the crucial research and conservation needs for these two under-studied and under-represented orders of mammals. Through regional-level planning, there is a greater likelihood of influence on policy, a higher probability of successful conservation interventions, and an increased likelihood of local stakeholder/community support.

We discuss these two case studies and the lessons learned from these two exercises. We close with recommendations on how to strengthen and formalize the species prioritization process.

### 2.2. Species Prioritization Requirements

Effective prioritization requires the use of multiple categories of variables, including biological, economic, and land use categories, current conservation status, and management

options. These include factors, such as the Red List status, known threats, protected area or KBA coverage, EDGE rank, genetic diversity, abundance information, trends, knowledge levels, researcher resources, funding availability, management potential, captive breeding potential, indigenous and local knowledge, land/resource conflicts, and confidence in effective governance.

The categories of variables and the specific variables within each category will, by necessity, vary across species. A conscious attempt, however, should be made to guarantee that the biological category is not overemphasized at the cost of others. Biological categories are easier to quantify, however, and other qualitative categories, such as the potential for local governance and management, can be very case specific and are more ordinal or descriptive variables than quantitative ones.

### 2.3. Specialist Group Prioritization

The ASG and SMSG exercises were based on a similar approach and had both overlapping and unique variables, resulting in a slightly different process for prioritization. Not all variables applied to both sets of species. The variables developed included a mix of biological and socio-political metrics, both theoretical and practical, and a mix of quantitative, qualitative (primarily rankings), and descriptive variables. The variables were arranged in a matrix as columns, and then the species were listed in rows and assessed against the variables. The framework can be applied and further developed on a country- and taxon-specific basis since local factors, such as political feasibility, availability of funds, and land tenure practicalities, will influence the prioritization process. As part of the expert discussion of the process, an attempt was made to weight variables, but this aspect is yet to be finalized; this process can be contentious and difficult to get right [27].

The ASG process was conceived in outline on completion of the Regional Action Plan [28] and developed further in 2018, following the Global Mammal Assessment. The aim was to supplement the Red List Assessment data with a finer-scale framework to inform the priorities for planning and action and channel the available resources to where they would be most effective. The structure of the five Red List criteria means that even within the same category of threat, a species may be increasing, stable, or declining, at different rates, and/or have a widely different population size, or be subject to a differing intensity of threats or levels of conservation action (see [30]). The variables used (Table 1) include the Red List category, evolutionary distinctiveness (expressed as the number of species in the genus); population size and trend; the number of locations; the proportion of the range inside protected and conserved areas; the presence and size of an ex situ population (which is important to safeguard against the complete extinction of a species and needed for future reinforcement or reintroductions); the level of conservation action; the scope, severity, and reversibility of threats, and feasibility criteria (security, governance, and cost). The reliability of population size and trend data was assessed as high, medium, or low. The existence or not of a species recovery plan was added to the framework later.

The SMSG exercise was carried out in Puebla, Mexico, in April 2018, using the model template previously developed by the ASG as a starting point. Participation in the exercise included four members of the core SMSG team (Lacher, Kennerley, Nicolette Roach, Shelby McCay), facilitated by Luis Carillo, a Mexican representative from the IUCN SSC Conservation Planning Specialist Group, and Mexican technical experts from across academia, government, NGOs, and zoos (Appendix A). The SMSG exercise was developed with 18 variables based upon the ASG list (Table 2) with a few substitutions, most significantly, the inclusion of the EDGE (Evolutionarily Distinct Globally Endangered, https://www.edgeofexistence.org/, accessed on 25 July 2022) ranking. A full species list of all SMSG species was first developed for the workshop, then trimmed to reflect only the globally threatened species, which were then divided up into groups for teams with the relevant experience and expertise to assess the data in the prioritization template (see Table S1 for an example). The teams reviewed 76 species in total during the workshop. The groups then reconvened to report back, refine, and agree on the information. For each

species, an 'Urgency' score and 'Feasibility' score were assigned, with a score of 1–3, with 3 being high. Species were initially ranked by Feasibility, then Urgency. We also considered if the conservation actions for the target species would also likely benefit other globally threatened small mammals in the habitat or region.

**Table 1.** ASG variable list and definitions.

| Variable | Definition |
|---|---|
| Red List | IUCN Red List category and criteria |
| Tax_Uniq | Higher taxon richness (measured as the number of species in the genus) |
| Pop_size | Count, estimate, or range of population |
| Data_qual | Observed, estimated, or inferred quality |
| Reliabil | ASG assessment of reliability of previous two rows (high, medium, low) |
| Trend_gen | Increasing, decreasing, stable, unknown |
| Trend_rate | As in the Red List (e.g., EN can be 51% to 79%) |
| Data_qual_2 | ASG assessment of previous two rows (high, medium, low) |
| Locations | Number of locations (as defined in the Red List categories and criteria) |
| PA_% | Percentage of population in protected areas (if use range as proxy add -R, e.g., 30-R) |
| Capt_no | Number in breeding programs |
| Capt_GD | Genetic diversity or #founders (high, medium, low) |
| Cons_Prog | Ongoing conservation programs (HML) |
| Thr_scope | Percent of the range impacted |
| Thr_trend | Increasing, decreasing, stable, unknown trend |
| Thr_revers | Reversibility of threats (high, medium, low) |
| Feas_secur | Feasibility of security action |
| Feas_gover | Feasibility of successful action due to governance issues (high, medium, low) |
| Feas_cost | Cost-effectiveness (logistics, access, travel) |

**Table 2.** SMSG variable list and definitions.

| Variable | Definition |
|---|---|
| Red List | IUCN Red List category |
| Red List criteria | IUCN Red List criteria from assessment |
| EDGE | EDGE ranking |
| Pop_size | Count, estimate, or range of population |
| Data quality | Observed, estimated, or inferred quality. |
| Trend | Increasing, decreasing, stable, unknown trend |
| Data quality | Assessment of previous two rows (high, medium, low) |
| Number locations | Number of locations (as defined in the Red List categories and criteria) |
| PA_coverage | Percentage of population in protected areas |
| In captivity | Yes/No, where |
| Captive Breed Prog | Number in breeding programs |
| Strong Cap Breed motive | Yes, No, Maybe |
| Invasive species | Any documented invasive species threats |
| Active mgmt. populations | Yes/No/where |
| Current threats | Yes/no, describe in comments |
| Thr_scope | Percent range impacted |
| Thr_trend | Increasing, decreasing, stable, unknown trend |
| Thr_revers | Reversibility of threats (high, medium, low) applied to main threats only |
| Comments | Relevant comments regarding any aspect |

## 3. Results

The ASG framework was applied to the 29 species assessed as globally threatened in the latest Global Mammal Assessment (Table 3). This framework clearly shows several priority species and areas. For example, three threatened species occurring in the Horn of Africa (*Gazella spekei*, *Ammodorcas clarkei*, *Dorcatragus megalotis*) have small population sizes, do not occur in any protected areas, and have no ex situ populations, yet the feasibility of effective conservation action is severely limited by a high level of insecurity and limited capacity.

**Table 3.** ASG prioritization template and metrics for the 29 key species. Peach heading color indicates Red List demographic, and evolutionary distinctiveness variables; pale green is conservation and threat factors; and pale blue variables are feasibility categories. Red indicates Critically Endangered, orange Endangered, and yellow Vulnerable under Red List Status. The pink highlighted cells indicated variables important for the highest priority species. * Under Population Trend D is decreasing and I is increasing. ** Under Recovery Plan the year indicates the date of publication of the action plan/conservation strategy/recovery plan and prep indicates that a plan or strategy is under development. Under Population size, ? indicates an uncertain estimate and ?? denotes no reliable estimate.

| | Red List Status | TaxU (# in Genus) | Pop Size | Pop Trend * | Data Quality | Locations | PCA % | PCA Effectiveness | Ex Situ | Recovery Plan ** | Cons Action | Thr Scope (%) | Thr Severity | Thr Irreversibility | Feas Insecurity | Feas Govern | Feas Cost |
|---|---|---|---|---|---|---|---|---|---|---|---|---|---|---|---|---|---|
| *Addax nasomaculatus* | CR | 1 | 50 | D | H | 2 | 0 | - | 5000 | 2017 | H | 100 | H | H | M | M/H | H |
| *Beatragus hunteri* | CR | 1 | 500 | D | H | 1 | 10 | M | 0 | 2021 | H | 100 | H | H | H | M | M |
| *Nanger dama* | CR | 3 | 50–100 | D | M | 3–4 | 80 | M | 1200 | 2019 | M | 100 | H | H | H | M/H | H |
| *Saiga tatarica* | CR | 1 | 1,300,000 | I | H | 5 | 25–50 | H | 400 | 2021 | H | 100 | L | L | L | L | L |
| *Tragelaphus buxtoni* | EN | 68 | 2000–4000 | D | M | 2–3 | 50–75 | M-H | 0 | | M | 50 | H | H | L | M | H |
| *Kobus megaceros* | EN | 4 | 4300 | D | L | 2 | 25–50 | L | 450 | | L | 100 | M | H | H | H | H |
| *Oryx beisa* | EN | 4 | 16–18,000 | D | M | >10 | 25–50 | M | 820 | | M | 80 | H | M | L/M | L/M | M |
| *Redunca fulvorufula* | EN | 4 | 12–16,000 | D | M | >10 | 25–50 | M-H | 21 | | L | 60 | H | M | L | L/M | M |
| *Eudorcas tilonura* | EN | 4 | 250? | D | L | 2 | >50 | L | 0 | prep | L | 100 | H | H | H | M | M |
| *Gazella leptoceros* | EN | 8 | 800–1000 | D | L | 2–3 | 10 | M | 80 | 2019 | L | 100 | H | H | M | L | M |
| *Gazella gazella* | EN | 8 | 3500 | D | H | >10 | 50–75 | H | 20 | | H | 40 | H | M | L | L | L |
| *Gazella spekei* | EN | 8 | ?? | D | L | 1 | 0 | - | 176 | | L | 100 | M/H | M | H | M/H | M/H |
| *Procapra przewalskii* | EN | 3 | 1300–1600 | I | H | 5 | 75–100 | H | 0 | 2004 | H | 100 | M | L | L | L | L |
| *Cephalophus jentinki* | EN | 16 | 3500 | D | L | 3 | 50–75 | M | 0 | | L | 100 | M/H | H | L | L | M/H |
| *Cephalophus spadix* | EN | 16 | 1500 | D | M | 3 | 50–75 | M | 0 | | L | 100 | H | H | L | L | M/H |
| *Tetracerus quadricornis* | VU | 1 | 10,000 | D | M | >10 | 25–50 | M | 127 | | L | 50 | M | M | L | L | L |
| *Tragelaphus derbianus* | VU | 68 | 12–14,000 | D | M | 4 | 50–75 | M | 44 | | M | 50 | H | M | L | L | M |
| *Oryx leucoryx* | VU | 4 | 1200 | I | H | 5–8 | 100 | H | 8000 | 2007 | H | 10 | L | L | L | L | L |
| *Ammodorcas clarkei* | VU | 1 | 4–5000 | D | L | 1 | 0 | - | 0 | | 0 | 100 | M? | M? | L | L | M |
| *Nanger soemmerringii* | VU | 3 | 6000–7500 | D | L | 4 | <25 | L | 42 | | L | 75 | H | M | M | M | M |
| *Eudorcas rufifrons* | VU | 4 | 12,000 | D | L | >10 | <25 | L | 26 | prep | L | 100 | H | H | M | L/M | H |
| *Gazella arabica* | VU | 8 | 10,000 | D | H | 8–9 | 90 | H | 80,000 | | H | 25 | L | L | L/M | L | L |
| *Gazella cuvieri* | VU | 8 | 2300–4500 | D | M | >10 | <25 | L-M | 37 | 2017 | M | 75 | M | M | L | L | L |
| *Gazella dorcas* | VU | 8 | 10,000 | D | M | >10 | <25 | L-M | 262 | prep | L | 80 | M/H | M | L/M | L/M | L |
| *Gazella marica* | VU | 8 | 3000 | D | H | 6 | 90 | H | 32,000 | | H | 100 | L | L | L | L | L |
| *Gazella subgutturosa* | VU | 8 | 40,000 | D | M | >10 | 25–50 | M | 404 | | M | 75 | M/H | M | L | L | L |
| *Dorcatragus megalotis* | VU | 1 | 7000 | D | L | 3 | <25 | M | 0 | | L | 100 | L/M | L | H | M/H | M/H |
| *Cephalophus adersi* | VU | 16 | 20,000 | D | M | 3 | >75 | L | 0 | | L | 100 | H | H | H | M | M/H |
| *Cephalophus zebra* | VU | 16 | 15,000 | D | L | 5 | 50–75 | M | 0 | | L | 100 | M/H | H | L | L | M/H |

The poor situation of antelopes in the Horn of Africa, portrayed here, led directly to a proposal for the development of a national antelope conservation strategy for the 30 species of antelopes in Ethiopia, in collaboration with the Ethiopian Wildlife Conservation Authority. This exercise utilizes a modified prioritization framework, adapted for use at national and regional levels, i.e., adding variables to show the global population's share of each species at national (Ethiopia) and regional (Horn of Africa) levels. Finalizing this exercise was delayed, first by the COVID epidemic and the associated travel restrictions and, subsequently, by an ongoing civil conflict. The framework can be further developed to cover antelope subspecies, Green Status of Species 'spatial units', or regional populations. It can also be updated to reflect changes in any of the variables, thus providing a real-time 'snapshot' of the global situation. For example, saiga antelope, *Saiga tatarica*, had

an estimated population size of 164,600–165,600 in January 2018 [31], but their numbers increased dramatically to >1.3 million by May 2022, indicating a change in recommended priorities. The data quality assessment highlights where more thorough work is needed to improve the accuracy of population estimates and extrapolations, and thus the reliability of the Red List assessments. The matrix also shows that several threatened species have no, or only a very small, ex situ (insurance) population, and this information can assist the decisions on establishing or reinforcing these (see [32]), though ex situ conservation is not recommended for some antelope species that are difficult to maintain in captivity.

The results of the Mexico prioritization exercise selected 19 CR species and ranked them based upon the consensus distillation of the templates into three broad categories: Feasibility (known distribution, investigator access, NGO capacity, ability to implement management, availability of capable researchers to do the project), Urgency (magnitude of current threats and trends in those threats), and Additional Species Benefitted (Table 4). One of the early priorities developed by the team members was to select those species for which the likelihood of defining and implementing conservation actions was high. Chasing projects with low feasibility was time and resources poorly spent; thus, we ranked the 19 species first on their feasibility. All 19 species were also scored by Urgency (magnitude of threats) and their benefits to other species that co-occurred.

**Table 4.** Results of the prioritization process for critically endangered Rodentia and Eulipotyphla species for Mexico, with the top 6 species selected, in bold text.

| Rank | Species | Feasibility Score | Urgency Score | Number of Other GT Species That Would Benefit |
|:---:|:---:|:---:|:---:|:---:|
| **1** | ***Xenomys nelsoni*** | **3** | **3** | **3** |
| **2** | ***Peromyscus guardia*** | **3** | **3** | **2** |
| **3** | ***Reithrodontomys spectabilis*** | **3** | **3** | **2** |
| **4** | ***Peromyscus slevini*** | **3** | **3** | **1** |
| **5** | ***Habromys schmidlyi*** | **3** | **2** | **1** |
| **6** | ***Habromys ixtlani*** | **3** | **1** | **1** |
| 7 | *Sorex sclateri* | 2 | 3 | 3 |
| 8 | *Geomys tropicalis* | 2 | 3 | 1 |
| 9 | *Orthogeomys lanius* | 2 | 3 | 1 |
| 10 | *Habromys chinanteco* | 2 | 3 | 1 |
| 11 | *Habromys delicatulus* | 2 | 3 | 1 |
| 12 | *Habromys lepturus* | 2 | 3 | 1 |
| 13 | *Habromys simulatus* | 2 | 3 | 1 |
| 14 | *Tylomys bullaris* | 2 | 3 | 1 |
| 15 | *Neotoma nelsoni* | 2 | 2 | 1 |
| 16 | *Peromyscus bullatus* | 2 | 2 | 1 |
| 17 | *Tylomys tumbalensis* | 1 | 3 | 3 |
| 18 | *Sorex stizodon* | 1 | 3 | 2 |
| 19 | *Heteromys nelsoni* | 1 | 3 | 1 |

All six species ranked highest in feasibility had highly restricted ranges. Three (*Peromyscus guardia*, *P. slevini*, *Reithrodontomys spectabilis*) are found on islands only, two (*Habromys ixtlani*, *H. schmidlyi*) have small ranges in the mountains, and *Xenomys nelsoni* is restricted to a strip of coastal lowland forest. They were viewed as having ranges in areas that were more likely to be amenable to conservation through heavy management. The three lowest feasibility scores went to three species (*Tylomys tumbalensis*, *Sorex stizodon*, and *Heteromys nelsoni*) due to poor knowledge, resulting from data only from a type locality for the first two species, and difficulty of access for the latter species.

Based on these scores, a matrix of conservation action plan ideas was developed for the top six species and circulated back to the team members with expertise in the relevant

species and habitats. After the workshop, a team of students from Texas A&M University developed a draft of the conservation action plans based on the available data. Two of these action plans (*Peromyscus guardia* and *Reithrodontomys spectabilis*) were used to develop proposals for the National Geographic Society Species Recovery grants; however, neither was successful—a recurring difficulty in supporting rodent conservation. Two other priority species (*Habromys chinanteco* and *Sorex sclateri*) are of interest to Rainforest Trust for possible conservation support. An additional benefit of using the templates was the development of an additional list of species for which the current Red List assessments needed either revision or reassessment (Appendix B). Two species on that list (*Dipodomys gravipes* and *Cynomys mexicanus*) have also received increased conservation interest, resulting in proposal development.

## 4. Discussion

The IUCN Red List assessments are focused on species, and the new IUCN Assess–Plan–Act framework has encouraged SSC Specialist groups to undertake planning and consider relevant activities beyond assessment. The Conservation Planning Specialist Group publishes guidelines [33] and species-level action plans, with the more recent documents including the A2P conceptual framework for multi-species conservation action planning [34]. The process encourages the inclusion of categories of information related to sites, habitats, threats, possible recovery plans, and assisted recovery, such as captive breeding and reintroduction. In our prioritization efforts, we included variables to cover this range of information.

A critical component is to define both the target and methodology upfront. For example, measures to conserve evolutionary processes and current patterns of diversity may conflict [12]. Variables should be clearly defined, transparent and repeatable. It is important to define all relevant variables at the beginning and include them in the initial analysis. The complexity can be reduced and simplified as the prioritization process progresses; however, the diversity of the information contributes to the development of priorities that include biological, ecological, and socio-economic data and constraints.

Both prioritization exercises were conducted prior to the advent of COVID-19. The ASG exercise was primarily expert-led and was intended to provide an updateable global summary of antelope priorities, though it can be adapted for use at a national level, and it will be deployed in planning workshops. The SMSG exercise involved in-person workshops. Although this greatly facilitated lively and open discussions, over the past several years, we have learned how to improve the level of interaction in virtual workshops. Despite this, we believe in-person workshops are preferable, especially when they are regionally focused and most participants are local, which means that travel costs and climate impacts are low. This allows for all local experts to come together and interact, meaning that the relative importance of the variables, and their interactions, can be discussed and evaluated. Maximizing local expertise is essential, in our view. The ASG framework was originally developed for use at a global level by ASG leadership and the SSC. Application at a regional or national level or within specific workshops will be completed in collaboration with the relevant government and academic experts, such as in the ongoing planning process for antelopes in Ethiopia. In Mexico, we were able to bring together almost all the small mammal experts from the region (Appendix A). This served to validate the outcome of the prioritization exercise since it was built on local expertise. This also established cooperation with local and national agencies, as they have been part of the process from the beginning.

The documenting of decisions has become an essential component of conservation science, as it allows for future adaptive management [35,36]. The process in both workshops required careful discussion and documentation of the weighting of variables, which is a critical aspect of the exercise, discussed previously by [27]. However, assigning relative weights to individual variables frequently entails arbitrary and subjective judgements and can be both challenging and contentious, as changing weights can result in very different overall priorities. Active discussion during the workshops contributed to a better

understanding of the weighting challenges and importance. A weighting scheme has not yet been finalized for the ASG framework. Having flexibility in applying the criteria to different species under many different levels of scientific knowledge of feasibility was critical in deriving the final priorities. There were several discussions about how to balance the decisions between a biological priority (e.g., closeness to extinction) with a practical priority (e.g., what can be done on the ground given the resources and logistics). In the Mexican assessment, this was incorporated into the "Feasibility Score" [37]. Some regions of the country presented very restricted opportunities to conduct additional field research or allow for the implementation of conservation activities due to economic, knowledge, or safety restrictions.

Some criteria were more difficult to apply to the prioritization than others. The ASG framework included a measure of taxonomic uniqueness (intrageneric richness) as a criterion in order to include considerations of phylogeny in the prioritization [38], and the Mexican framework included the EDGE score of each species for which it was available [39]. Although the Mexican assessment intended to use the EDGE score as a metric, logistical and feasibility questions consistently trumped EDGE as a selection criterion. In addition, there were some disagreements concerning the mismatches between the IUCN taxonomy and the perspectives of the Mexican team. Ranking, using Alliance for Zero Extinction sites as a priority, had mixed results, where it was valuable for some island-restricted species currently unprotected or under threat. Similar to EDGE, it was less the fact that a species was at an AZE site than other logistical, urgency, and feasibility issues. This again emphasizes the importance of gaining the participation of local expertise rather than relying on a quantitative formula for rating priorities. The potential for captive breeding programs did influence some of the prioritization for both specialist groups. In the SMSG, the possibility of captive breeding was a key factor for *Peromyscus guardia,* given the possibility of only a few remaining individuals on the island. This was also an important consideration for the ASG with *Addax nasomaculatus* and *Nanger dama,* which are close to extinction in the wild but have large ex situ populations that are providing stock for reintroduction and demographic or genetic reinforcement.

The framework we jointly developed in theory can be applied to other taxa, certainly all mammals and perhaps most vertebrates. The criteria would need to be adjusted for prioritization exercises involving plants and insects. The SMSG and ASG also plan to use this method for upcoming assessments in other key regions and countries, in a Sulawesi-focused assessment by the SMSG, and in an ongoing assessment in Ethiopia by the ASG.

### 5. Conclusions

(1)　The Antelope Specialist Group and the Small Mammal Specialist Group recently conducted species prioritization workshops to guide conservation actions.
(2)　We used a matrix of quantitative and qualitative metrics in both cases, covering biological, social, and economic considerations.
(3)　Certain metrics assumed higher weights as the process moved forward and were influenced by factors, such as feasibility, urgency, and broader conservation impacts.
(4)　For viable priorities to be developed, there must be dominant participation of experts from the regions involved.

**Supplementary Materials:** The following supporting information can be downloaded at: https://www.mdpi.com/article/10.3390/d14090704/s1, Table S1: SMSG sample template.

**Author Contributions:** Conceptualization: T.E.L.J., D.M., R.J.K. and R.P.Y.; Formal analysis: T.E.L.J., D.M., R.J.K. and C.R.; Methodology: T.E.L.J., D.M., R.J.K. and C.R.; Project administration: R.P.Y.; Writing—original draft: T.E.L.J., D.M., R.J.K. and C.R.; Writing—review and editing: T.E.L.J., D.M., R.J.K. and C.R. All authors will be informed about each step of manuscript processing, including submission, revision, revision reminder, etc., via emails from our system or assigned Assistant Editor. All authors have read and agreed to the published version of the manuscript.

**Funding:** The Mexico SMSG workshop received logistical and financial support from AfriCam Safari, Jacksonville Zoo and Gardens, Houston Zoo and Nashville Zoo. Texas A&M University provided travel support.

**Institutional Review Board Statement:** Not applicable.

**Data Availability Statement:** Summary reports of the workshops and data are available through the authors, DM for the IUCN SSC Antelope Specialist group and TL or RK for the IUCN SCC Small Mammal Specialist Group.

**Acknowledgments:** The SMSG would like to thank the following for their support for the Mexico SMSG workshop: AfriCam Safari, Jacksonville Zoo and Gardens, Houston Zoo and Nashville Zoo. Additional general support to the IUCN SSC Small Mammal Specialist Group was from Brevard Zoo, Disney's Animal Kingdom, Greensboro Science Center, Lincoln Park Zoo, Mesker Park Zoo, Milwaukee County Zoo, Philadelphia Zoo, Smithsonian's National Zoo, Zoo Atlanta, and Zoo New England. Texas A&M University provided some travel support.

**Conflicts of Interest:** The authors declare no conflict of interest.

## Appendix A

**Table A1.** Participants in SMSG Mexico workshop and workshop follow-up and institutional ties at the time of the workshop.

| Name | Organization |
| --- | --- |
| Frank Carlos Camacho | Director General, AfriCam Safari |
| Luis Martínez | AfriCam Safari |
| Sergio Ticul Álvarez-Castañeda | Centro de Investigaciones Biológicas del Noroeste, La Paz, Baja California Sur, México |
| Rafael Ramírez | CONABIO México |
| Luis Carrillo | Conservation Planning Specialist Group—Mexico |
| Luis Verde Arregoitia | Instituto de Ciencias Ambientales y Evolutivas, Universidad Austral de Chile |
| Joaquín Arroyo-Cabrales | Instituto Nacional Antropología e Historia |
| Alfredo Cuarón | SACBÉ—Servicios Ambientales, Conservación Biológica y Educación A.C. |
| Ros Kennerley | SMSG Durrell Wildlife Conservation Trust |
| Tom Lacher | SMSG Texas A&M University |
| Nikki Roach | SMSG Texas A&M University |
| Shelby McCay | SMSG Texas A&M University |
| Víctor Sánchez Cordero | Universidad Nacional Autónoma de México (UNAM), Instituto de Biología |
| Lázaro Guevara | UNAM, Instituto de Biología |
| Francisco Botello | UNAM, Instituto de Biología |
| Ella Vázquez-Domínguez | UNAM, Instituto de Ecología |
| Gerardo Ceballos | UNAM, Instituto de Ecología |
| Marcial Quiroga-Carmona | Universidad Austral de Chile (PhD Candidate) |
| David Obed Vázquez Ruiz | Laboratorio de Ecología y Conservación de Fauna Silvestre, UNAM |

## Appendix B

**Table A2.** Species Red List accounts to be reviewed because of new information gathered during the workshop.

| Species Name | Current Red List Category and Criteria | Species Name | Current Red List Category and Criteria |
| --- | --- | --- | --- |
| *Cryptotis nelsoni* | CR B1ab(i,ii,iii) | *Peromyscus ochraventer* | VU B1ab(iii) |
| *Dasyprocta mexicana* | CR A2c | *Reithrodontomys hirsutus* | VU B1ab (iii) |
| *Dipodomys gravipes* | CR D | *Sigmodon alleni* | VU A2c + 3c + 4c |
| *Habromys simulatus* | CR C2a(i,ii) | *Sorex macrodon* | VU B1ab(iii) |
| *Neotoma nelsoni* | CR B1ab(iii) | *Sorex milleri* | VU B1ab(iii) |
| *Peromyscus caniceps* | CR B1ab(v) | *Microtus quasiater* | NT |
| *Peromyscus dickeyi* | CR B1ac(iv)+2ac(iv) | *Neotoma phenax* | NT |

**Table A2.** *Cont.*

| Species Name | Current Red List Category and Criteria | Species Name | Current Red List Category and Criteria |
|---|---|---|---|
| *Peromysucs guardia* | CR(PE) B2ab(iv,v) | *Peromyscus polius* | NT |
| *Peromysucs interparietalis* | CR B1ab(v) | *Rheomys thomasi* | NT |
| *Peromyscus pseudocrinitus* | CR B1ab | *Chaetodipus lineatus* | DD |
| *Peromyscus slevini* | CR B1ab(v) | *Cryptotis alticola* | DD |
| *Tylomys tumbalensis* | CR B1ab(iii,v) | *Cryptotis peregrina* | DD |
| *Cryptotis griseoventris* | EN B1ab(i,iii) | *Cryptotis tropicalis* | DD |
| *Cynomys mexicanus* | EN B1ab(i,ii,iii,iv) | *Neotoma insularis* | DD |
| *Habromys delicatulus* | EN B1ab(i,ii,iii)+2ab(i,ii,iii) | *Orthogeomys cuniculus* | DD |
| *Heteromys nelsoni* | EN B1ab(I,ii,iii,v) | *Peromyscus furvus* | DD |
| *Megadontomys nelsoni* | EN B2ab(iii) | *Peromyscus sagax* | DD |
| *Microtus umbrosus* | EN B1ab(i,ii,iii)+2ab(i,ii,iii) | *Reithrodontomys burti* | DD |
| *Nelsonia goldmani* | EN B1ab(iii) | *Chaetodipus pernix* | LC |
| *Peromyscus melanocarpus* | EN B1ab(iii) | *Chaetodipos siccus* | LC |
| *Reithrodontomys tenuirostris* | EN B1ab(i,iii) | *Neotomodon alstoni* | LC |
| *Rheomys mexicanus* | EN B1ab(iii) | *Peromyscus gymnotis* | LC |
| *Xenomys nelsoni* | EN B1ab(iii) | *Peromyscus hooperi* | LC |
| *Xerospermophilus perotensis* | EN B1ab(iii) | *Peromyscus schmidlyi* | LC |
| *Zygogeomys trichopus* | EN B1ab(iii,v) | *Tamias durangae* | LC |
| *Handleyomys chapmani* | VU B2ab(ii,iii) | | |
| *Handleyomys rhabdops* | VU B1ab(iii) | | |

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
