# Peer review of "Tools and Metrics for Species Prioritization for Conservation Planning and Action: Case Studies for Antelopes and Small Mammals"

_diversity, doi:10.3390/d14090704_

Round 1
Reviewer 1 Report
The manuscript is well-written. It would be better to check throughout the manuscript to avoid typo errors.
Author Response
Reviewer 1 requested only that we proof read the manuscript carefully, which we have done.
Reviewer 2 Report
see attachment.

Author Response
First I wish to apologize, but during the movement of the manuscript back and forth among the authors, the Track Changes were accepted. There were, however, few edits requested and the lines where those changes were made are indicated below.
- Please add a table in the Materials and Methods to provide a general summary of the comparison between the variables and matrices used in this paper and those used in previous references.
Given how varied the papers were in reporting criteria and variables, a Table just didn't work out, so we added additional information in the text to clarify this and provide some additional detail. Those changes are in lines:
63-69, 79-80, 84, and 113-116.
2. Both ASG and SMSG workshops were conducted in 2018, which were 4 years ago, please provide some update of the decisions or actions made after these two workshops. For example, in Line 210-213, it stated “The poor situation of
antelopes in the Horn of Africa portrayed here led directly to a proposal for
development of a national antelope conservation strategy for the 30 species of
antelopes in Ethiopia,….” Any recent development of this action? Same as to the
SMSG evaluation result, any update on the implementation of the conservation
action plans which were developed for the six high priority species?
We have added additional information related to follow-up actions in the text in the floowing lines:
205-209, 239-249.
Finally all minor grammatical errors were corrected.

Reviewer 3 Report
Dear authors,
Thank you for the opportunity to read your manuscript. I have no doubt that will be a huge contribution for species evaluation from IUCN groups. In summary, the manuscript for me has no flaws and all topics rise their contents. The English is well written, and the tables are meaningful. The discussion and conclusion are eloquent and brings good discussion about the issue. The supplementary data is ok (Excel table), as well the appendix. The references are also actual and has classics papers about the species evaluation form IUCN. For the record, I would like to see this manuscript published and the methodology abroad used from scientist and IUCN groups.
Best wishes
Author Response
We thank reviewer 3 for the very positive comments, and the reviewer suggested no changes be made.